# Aflatoxin B1 Contamination Association with the Seed Coat Biochemical Marker Polyphenol in Peanuts Under Intermittent Drought

**DOI:** 10.3390/jof10120850

**Published:** 2024-12-10

**Authors:** Maman Moutari Aminou, Hamidou Falalou, Harou Abdou, Venugopal Mendu

**Affiliations:** 1Department of Biology, Faculty of Sciences, Abdou Moumouni University of Niamey, Niamey P.O. Box 10662, Niger; amamanmoutari@gmail.com; 2International Crops Research Institute for the Semi-Arid Tropics (ICRISAT), Niamey P.O. Box 12404, Niger; 3Department of Biology, Faculty of Sciences, Adré Salifou University, Zinder P.O. Box 656, Niger; aharousouley@gmail.com; 4Department of Agriculture, Agribusiness, and Environmental Sciences, Texas A&M University, 700 University Blvd, MSC 228, Kingsville, TX 78363, USA; venugopal.mendu@tamuk.edu

**Keywords:** peanut, intermittent drought, seed coat, aflatoxin

## Abstract

Aflatoxin B1 (AFB1) contamination (AC) increases as the severity of drought stress increases in peanuts. Identifying drought-tolerant (DT) genotypes with resistance to *Aspergillus flavus* colonization and/or infection may aid in developing peanuts resistant to aflatoxin contamination in the semi-arid tropics. The goal of this study is to identify DT genotypes with seed coat biochemical resistance to *A. flavus* infestation and aflatoxin contamination. Experiments were carried out at ICRISAT Sahelian Center; fifty-five genotypes were assessed under adjacent intermittent water-stressed (WS) conditions imposed from the 60th day after sowing to the maturity date and well-watered (WW) conditions in an alpha lattice design with two factors. The yield and its components, the incidence of *A. flavus* colonization, aflatoxin contamination, and seed coat total polyphenol (SCTPP) were investigated. Our findings show that the water deficit reduced the pod yield, seed yield, and haulm yield by up to 19.49%, 27.24%, and 22.07%, respectively, while it increased the number of immature pods per plant (IMPN) and the aflatoxin contamination by up to 67.16% and 54.95%, respectively. The drought tolerant genotypes ICG 2106, ICG 311, ICG 4684, ICG 4543, and ICG 1415 maintained a high yield, small number of IMPN under WS and low aflatoxin content variation between WW and WS. Our findings revealed that in the drought-tolerant genotypes ICG 1415, ICG 2106, ICG 311, ICG 4684, and ICG 4543, there was a significant relationship between the aflatoxin resistance and the seed coat total polyphenol under the two water treatments (r^2^ = 0.80; r^2^ = 0.82). This suggests that these drought-tolerant genotypes kept their seed coat intact and minimized the aflatoxin contamination under an intermittent water deficit.

## 1. Introduction

Peanut (*Arachis hypogaea* L.) is the second most important cash crop in Niger, where it is produced in five of eight regions of the country. It is called a ‘women’s crop’ because of the extensive involvement of women in peanut production and processing. Peanut pods grow underground, and their development is directly influenced by the water conditions of the surrounding soil [1]. As an underground crop, the pods are subjected to continuous risk of direct contact with populations of aflatoxigenic aspergilli in the soil [2]. Furthermore, given the ubiquitous nature of *Aspergillus flavus*, it is almost impossible to eliminate the exposure of peanut to the fungus [3]. *A. flavus* is an opportunistic fungal pathogen of crops, predominantly maize, peanuts, and cotton, characterized by a high potential for aflatoxin production [4]. Aflatoxins are potent, highly toxic secondary metabolites that can compromise food and feed security and cause severe health issues [5,6].

Aflatoxin contamination is the most important quality problem in peanuts throughout the world as it is related to serious health problems in humans as well as in livestock [7,8,9,10,11]. Peanut crops grown in Sahel often experience water deficits during the pod-filling phase, which usually coincides with the end of the rainy season [12,13,14]. In this zone, peanut production is often affected by intermittent drought, which is an episodic water deficit during plant growth [15]. Previous studies in Niger demonstrated that drought stress for fewer than ten days was enough to cause significant aflatoxin B1 contamination in the field [16,17,18]. However, an increased duration of terminal drought and temperature are major factors determining the level of aflatoxin B1 contamination [16].

Precisely, at the pod-filling stage, drought causes plant stress and leads to lower phytoalexin production [19]; then, *A. flavus* can colonize peanut pods prior to harvest, and contamination with aflatoxin is more severe under terminal drought conditions [14,20,21]. Indeed, drought can cause cracking in seed coats, which, in turn, permits the ingress of *A. flavus* germinating spores and hyphae into embryonic tissues [22]. Recent studies have observed that varieties subjected to drought during late growth stages had different levels of resistance to aflatoxin contamination [20]. The different levels of resistance are linked firstly to drought-tolerance mechanisms, by escape, tolerance, or avoidance, and may impact the ability of genotypes to minimize aflatoxin production by maintaining kernel water activities, allowing phytoalexin production [23]. Phytoalexins, which are antimicrobial agents produced by plants, are known to inhibit the production of aflatoxins in plants after colonization. Wound-induced stilbene phytoalexins in peanut kernels inhibit the germination of spores and hyphal extension of *A. flavus* [24]. Secondly, the seed coats and outer shells play vital roles in protecting the seeds from mechanical damage, pest infestation, and harsh weather conditions [25]. Then, a combined approach of host resistance followed by pre- and post-harvest management practices is required [26].

Aflatoxin contamination can be minimized by adopting certain cultural, produce-handling, and storage practices [23,27]. However, these practices are not widely adopted, particularly by small-scale farmers in the developing countries which contribute to about 60% of the world peanut production. One option to reduce aflatoxin contamination in peanut plants is the use of cultivars resistant to seed infestation by *Aspergillus flavus* and/or cultivars resistant to aflatoxin production. Researchers have suggested that drought-tolerant genotypes may possess some degree of tolerance to aflatoxin contamination, and they have argued that drought-tolerance traits in peanuts may have the potential to be used as indirect selection criteria for resistance to pre-harvest aflatoxin contamination [28]. A recent study [10] reported that the wide variation in the incidence and severity of *A. flavus* colonization and aflatoxin content between genotypes can be attributed to the biochemical compound variability in the tested seeds. It was observed that differences in mycelial growth surface coverage could probably be attributed to differences in the physical and chemical features of the seed coat, pod shell thickness, and reticulation [29]. It was recently demonstrated that the seed coat acts as a physical and biochemical barrier against *A. flavus* infection [24,30]. To develop peanut cultivars resistant to *A*. *flavus* infection with reduced aflatoxin contamination, there is a possible defense mechanism at three stages: prevention of fungal infestation in the pericarp; resistance to seed contamination and colonization (seed coat); and resistance to aflatoxin production (cotyledons) [31]. Using cultivars resistant to kernel infection by *A*. *flavus* is also one of the promising ways to reduce aflatoxin contamination. The objective of this study was to identify the drought-tolerant genotypes resistant to *A. flavus* colonization and to aflatoxin contamination. The specific objectives are to (1) evaluate peanut genotypes under an intermittent water deficit to identify the drought-tolerant genotypes, (2) identify by in vitro seed colonization (IVSC) the seed coat biochemical marker polyphenol in genotypes resistant to *A. flavus* colonization, and (3) investigate the potential relationship between drought-tolerance, seed coat resistance to *A. flavus* infection, and aflatoxin contamination.

## 2. Materials and Methods

### 2.1. Plant Materials, Temperature, and Relative Humidity

Fifty-five genotypes from the ICRISAT peanut mini core collection provided by the ICRISAT Niger gene bank were selected according to their contrasting levels of aflatoxin content [32] and the diversity of the seed coat colors. The materials include the varieties 55-437 and J11 considered as resistant to pre-harvest aflatoxin contamination, and JL24 and Fleur11 considered as susceptible [33]. Two experiments were conducted (August to December) in year 1 and year 2 under field conditions at the International Crop Research Institute for the Semi-Arid Tropics (ICRISAT) Sahelian Centre (ISC) in Sadoré (45 km south of Niamey, Niger, 13° N, 2° E). During the crop growing period, the maximum (Max) and minimum (Min) air temperatures and the relative humidity (RH) were recorded daily from a meteorological station located close to the experimental field. The Max and Min temperatures varied from 26.4 °C to 41.2 °C in year 1 and from 29.01 °C to 39.30 °C in year 1, while the RH varied from 29.9% to 68.6% in year 2 and from 40.0% to 62.23% in year 2. The total water received from rainfall and irrigation was 590 mm^3^ and 470 mm^3^, respectively, in the well-watered (WW) and intermittent water-stressed (WS) treatment conditions in year 1 and 645 mm^3^ and 505 mm^3^ in year 2.

### 2.2. Experimental Design and Water Treatment

The experimental design was an alpha lattice with two factors: water treatment as the main factor, and genotypes (55) as sub-factors randomized in each water treatment with four replications. The spacing between the water treatments was 6 m and 2 m between replications. Within each repetition, there were 55 elementary plots with an area of 2 m × 1 m = 2 m^2^ spaced 1 m apart. The plots subject to WS were watered (30 mm^3^ per irrigation) like those of the WW treatment until the 60th day after sowing (start of the seed-filling stage) when the water stress was imposed. The WS treatment consisted of skipping irrigation of WS plants until the majority of the stressed plants showed clear wilting symptoms before watering and then skipping irrigation again. This cycle continued until the pods’ maturity.

### 2.3. Crop Management

In each experiment, the plot area measured 2 m^2^ (2 m × 1 m) including two rows of 2 m in length. The row spacing was 0.5 m and each row had ten hills. Three seeds were sown by hand in each hill at 3 cm deep after receiving an irrigation of 30 mm using a linear-moving irrigation system (Valmont Irrigation Inc., Valley, NE, USA). Two and three weeks after sowing, the plants were thinned to two and one plant(s) per hill, respectively. The plots were subsequently fertilized with 150 kg ha^−1^ N-P_2_O_5_-K_2_O and irrigated with 30 mm of water. The fields were kept free from weeds by manual cultivation, and regular spraying of decis (deltaméthrin 12 CE), Emacot 050WG (Emamectine benzoate 50 g/kg), and Benji controlled insect pests.

### 2.4. Measurements and Data Collection

#### Yield and Its Components

After harvesting by hand, the plants of each plot (1 m^2^) were collected to determine the haulm yield, the pod yield, and the seed yield. The central plants of each plot were used to determine the pod number per plant, the number of immature pods per plant, and the seed number per plant.

### 2.5. In-Vitro Seed Colonization Assay 

The in vitro seed colonization (IVSC) test was carried out in the laboratory using ten, fifteen, and twenty-five seeds per Petri dish. For each genotype, seeds without apparent damage on the coat were selected. Each Petri dish was considered as a replication using a modified method of [34]. The *A. flavus* strains were collected from the sorghum seeds exposed under laboratory conditions (25–32 °C) during seven days before the purification and multiplication on Potato Dextrose Agar (PDA) and the inoculum preparation [34]. The *A. flavus* colonies were collected based on morphological characteristics and homogenized by successive culture on PDA medium. The inoculum (conidial suspension) was prepared using the spores of *A. flavus*. The hemocytometer was used to estimate the spore concentration and was adjusted to a final concentration of 1.9 × 10^6^ mL^−1^. After the preparation, 500 µL of spore suspension was inoculated into each Petri dish before incubation at 28 °C for seven days in a dark room. The observation was recorded as the percent of colonization after incubation. Individual seeds were scored for surface colonization by *A. flavus* and for colonization severity using the following rating scale: 1 ≤ 5% seed surface colonized with scanty mycelial growth and no sporulation; 2 = 5–25% seed surface colonized with good mycelial growth and scanty sporulation; 3 = 26–50% seed surface colonized with good mycelial growth and good sporulation; 4 ≥ 50% seed surface colonized with heavy sporulation [34]. *A. flavus* colonization incidence determination (S) proceeds as follows:$$S=\frac{Seeds\;colonized}{Total\;seeds}\times 100$$

### 2.6. Seed Coat Total Polyphenol Extraction and Quantification

Extraction: The undamaged and matured seeds of all the studied genotypes were collected after drying. The seed coats of each genotype were removed from the embryo before grinding into powder. After grinding, 0.1 g of the seed coat powder of each genotype was introduced in the tube and 10 mL of methanol (50%) added before warming at 77 °C for 1 h.

Spectrometric Assay: The total polyphenol was determined using FCR (Folin–Ciocalteu reagent) [35]. The absorbance was estimated at 750 nm using a spectrophotometer (6715 UV/Vis JENWAY, Essex, UK). The calibration curve was plotted using 0–6 mL of solutions of 0.05 mg/mL of tannic acid concentration. The absorbance was measured to determine the content of total polyphenols using the following formula: C = (C1 × V)/W with C being the content of total polyphenols expressed in mg equivalent tannic acid g^−1^ of dry matter, C1 being the concentration of tannic acid established from the calibration curve in mg/L, V being the volume of extract in L, and W the quantity of seed coat powder.

### 2.7. Genotypes Aflatoxin Content Quantification

The aflatoxin B1 (AFB1) concentration in the seeds of fifty-five genotypes collected under well-watered treatment and water-stressed treatment was estimated by an Enzyme-Linked Immunosorbent Assay (ELISA). In each trial (year 1 and year 2), 100 g of seeds was collected in the two water regimes’ plots. Twenty grams of the fine powder was used for extracting aflatoxin B1 by dissolving in methanol 70% (*v*/*v*) containing 0.5% (*w*/*v*) KCl and homogenizing. The mix was filtered through Whatman No. 1 filter paper. The filtrate was diluted 1:15 with methanol and used in duplicate to estimate aflatoxin concentration by indirect competitive ELISA essentially as described by [36].

The ELISA kit was based on five principal steps: (1) coating antibodies, (2) blocking the reaction, (3) competition, (4) enzyme conjugation, and (5) substrate. Coating antibodies: In total, 15 mL of carbonate buffer (coating buffer) + 1.5 µL of aflatoxin B1 bovine serum albumin (AFB1–BSA) conjugate (1 mg/1 mL) was mixed well and 150 µL was distributed into each well of the ELISA plate. The ELISA plate was incubated at 37 °C for 45 min to 1 h before washing the plate with phosphate-buffered saline Tween 20 (PBST) 3–4 times. Blocking step: In total, 150 µL of 0.02% BSA was distributed to each well, incubated at 37 °C for 30 min, and the buffer discarded. The healthy groundnut sample (HGN) was prepared using the seeds of the aflatoxin-resistant control variety (J11). The mix was prepared using 1 mL of J11 sample + 9 mL (0.02% BSA). After the blocking step, 100 mL of HGN solution was added in the Wells B3 and C3 to B11 and C11. The standard polyclonal antibodies (100 µL/mL) mixture was prepared using 1 µL of the polyclonal antibodies in 6 mL of 0.02% BSA. After the mixing, 50 µL was added in each well and incubated at 37° C for 45 mn. After the incubation, the ELISA plate was washed 4 times.

The Enzyme-Linked antibodies (secondary antibody) was prepared with 1 µL of this antibody and 4 mL of 0.02% BSA solution. After the mixing, 150 µL was added in each well. The ELISA plate was incubated at 37 °C for 45 mn before washing 4 times. The substrate was prepared using 10 mL of 10% dimetylamine + 5 mg of p-nitrophenyl phosphate (pNPP) and dissolved. After shaking, 150 µL of the mix was added in each well and incubated in the dark for 20 mn. The optical density (OD) value was recorded at 405 nm using an ELISA reader. An Excel spreadsheet on the computer was used to draw a regression curve, and regression equation values were used to estimate the aflatoxin in each sample (www.icrisat.org/aflatoxin, accessed on 18 June 2023).

### 2.8. Analysis

The data used were the means of the two years of experiments. GENSTAT 14th edition (VSN international Ltd., Hemel Hempstead, UK) was used to perform the Shapiro–Wilk normality test before an analysis of variance (ANOVA) to assess the effects of genotype (G), water regime (W), and their interactions for the different measured traits. Multivariate analysis was performed using Minitab 16th edition to observe the groups of genotypes under well-watered and water-stressed treatments. Microsoft Office Excel 2013 Software (Microsoft Corp., Redmond, WA, USA) was used to investigate linear correlations between the yields, aflatoxin content, and seed coat total polyphenol content. In addition, tables and figures were produced using Excel version 2013.

## 3. Results

### 3.1. Yields and Yield Components

ANOVA revealed a significant genotype–water-treatment interaction (G × Wtrt) for all yield traits (Table 1), indicating that these parameters varied depending on the water treatments. Water deficit-imposed severity was observed mostly in the seed number per plant and seed yield rather than in the pod number per plant, pod yield, and haulm yield (Table 2).

Under well-watered conditions, the adequately irrigated plants showed a good growth of overground biomass (Figure 1A) without wilting symptoms. However, under water-stressed treatment, the plants showed symptoms of wilting on the leaves and stems (Figure 1B).

Under well-watered treatment, pods without shell damage were observed (Figure 2B), while under water-stressed treatment several types of damage were caused by water deficit on the pods. The damage of the pod shell structure led to fissures of the pod, disfigured shapes, spots, and the pods rotting (Figure 2B).

Seeds free of damage were observed under well-watered treatment (Figure 3A), while under water-stressed treatment the seeds showed several types of the damage such as discolored seed coats, shrunken seeds, decayed seeds, seeds covered by spots, and seeds with a small size (Figure 3B).

Under well-watered treatment, the highest pod number per plant (PNP) and seed number per plant (SNP) were observed in the genotypes ICG 10950, ICG 11322, ICG 4598, ICG 1415, ICG 1519, ICG 332, ICG 5609, ICG 6407, ICG 3992, and ICG 12235 (Figure 4). For pod yield (PY), seed yield (SY), and haulm yield (HY) under WW, the genotypes ICG 12697, ICG 12879, ICG 2019, ICG 4729, ICG 4543, ICG 4750, ICG 5195, ICG 5609, ICG 6407, and ICG 6703 showed the highest performance (Figure 4). Under WW treatment, the ascending hierarchical classification grouped the genotypes into three groups (Figure 4). Group 1 (G1) includes 27 genotypes, which have a low pod number per plant (PN) compared to the two others groups, a low immature pod number per plant (IMPN), and a high seed yield (SY); Group 2 (G2) consists of 20 genotypes which have high pod number per plant (PN), low immature pod number per plant (IMPN), high seed number per plant (SNP), a good haulm yield, and a high seed yield (SY); while Group 3 (G3) includes 8 genotypes which have a high pod number per plant (PNP), high immature pod number per plant (IMPN), high pod yield, high seed yield, and low haulm yield.

Under WS, the genotypes ICG 10950, ICG 1415, ICG 4598, ICG 5609, and ICG 6407 showed the highest pods, seeds, and a small number of immature pods (Figure 5). In addition, ICG 12879, ICG 14523, ICG 2019, and ICG 5195 showed high pod and seed numbers per plant. Our results showed that among the ten best-performing genotypes for pod yield, seed yield, and haulm yield under WS, apart from the top ten best performing genotypes, five others performed well under the well-watered treatment. These genotypes were ICG 12697, ICG 12879, ICG 2019, ICG 4729, and ICG 6703. In addition, ICG 4598, ICG 4684, ICG 2106, ICG 12988, and ICGIL 11114 also showed a good performance under WS. Therefore, the pod and seed yields of the genotypes ICG 12235, ICG 1142, ICG 14523, ICG 14106, ICG 14630, ICGIL 11125, and ICGIL17108 were more severely affected by the intermittent water stress during the seed-filling phase by showing the highest seed yield loss (Appendix A).

Under water stress, the ascending hierarchical classification grouped the genotypes into three groups (Figure 5). Group 1 (G1) includes 29 genotypes having a high pod number per plant (PN), low immature pod number per plant (IMPN), high seed number per plant (SNP), high haulm yield (HY), high pod yield (PY), and high seed yield (SY); Group 2 (G2) contains 25 genotypes having a low pod number per plant (PN), low immature pod number per plant (IMPN), high haulm yield (HY), low seed yield (SY), and a low pod yield; and Group 3 (G3) contains 1 genotype which has a high pod number per plant (PNP), high immature pod number per plant (IMPN), high pod yield, low seed yield, and good haulm yield (HY).

### 3.2. Intermittent Water Deficit (WS) Effect on Pre-Harvest Aflatoxin B1 Contamination (AC)

An analysis of variance revealed a significant genotype–water-treatment interaction (G × Wtrt) for aflatoxin contamination under well-watered and water-stressed treatments (Table 3). Under WW conditions, the lowest aflatoxin content was observed in five genotypes, ICG 12988 (0.28 µg kg^−1^), ICGIL 17108 (0.30 µg kg^−1^), ICG 5195 (0.40 µg kg^−1^), ICGIL 11110 (0.46 µg kg^−1^), and ICG 311 (0.48 µg kg^−1^), while ICG 332 (6.80 µg kg^−1^), ICG 11542 (5.56 µg kg^−1^), Fleur11 (4.01 µg kg^−1^), and ICGIL11114 (3.75 µg kg^−1^) showed the highest aflatoxin content under WW treatment (Figure 6). Intermittent water stress (WS) imposed during the seed-filling phase increased aflatoxin contamination by up to 67.16% (Figure 7). Under water deficit, the highest aflatoxin content was observed in ICG 332 (19.44 µg kg^−1^), ICG 6703 (18.65 µg kg^−1^), ICG 4764 (13.03 µg kg^−1^), JL24 (11.37 µg kg^−1^), and ICG 11542 (10.88 µg kg^−1^) (Figure 7). These genotypes were revealed to be more sensitive to aflatoxin contamination under drought, while ICG 311 (0.83 µg kg^−1^), ICG 2119 (0.90 µg kg^−1^), ICGIL 17108 (1.6 µg kg^−1^), ICG 4684 (1.93 µg kg^−1^), and 55-437 (2.15 µg kg^−1^) showed the lowest aflatoxin content under drought conditions. These genotypes were resistant to aflatoxin contamination (AC) under WS (Figure 7).

### 3.3. Seeds Colonization and Total Polyphenol Content of Seed Coat

ANOVA revealed a significant (*p* < 0.001) genotypic difference for the *A. flavus* colonization (AFC) rating. The total polyphenol content of the seed coat also showed a significant genotypic variation (*p* < 0.001) (Table 4). The two genotypes ICG 12988 (18%) and ICG 311 (19%) showed *A. flavus* infestation as well as the resistant control 55-437 (21%). These genotypes with 55-437 were the least infested (rate < 25%). This indicated their resistance to *A. flavus* colonization (Figure 8). They were followed by ICG 4543 (28.33%), ICG 11542 (28.33%), J11 (40%), and ICG 76 (42.5%) that showed an *A. flavus* infestation rate of less than 50%. The remaining genotypes showed an *A. flavus* infestation rating of more than 50%, indicating a high susceptibility to *A. flavus* colonization (Table 4). The highest total polyphenol content of the seed coat was observed in ICGIL 17108 (62.96%), ICG 3027 (57.50%), ICG 14630 (56.95%), ICG 12235 (56.67%), and ICGIL11125 (56.30%). All the tested genotypes showed more than 50% *A. flavus* colonization, while the lowest seed coat %TPP content was shown by ICG 532 (42.74%), ICG 1415 (43.68%), ICG 5609 (43.74%), and ICG 11542 (43.90%) (Table 5). Only ICG 11542 showed less than 50% *A. flavus* colonization.

From Figure 8, the genotypes ICG 311 (yellow) and ICG 12988 (yellow) showed the lowest incidences of *A. flavus* colonization, and 55-437 (yellow) was a resistant control. The genotypes ICG 12235 (red) and ICG 2119 (red) showed seed surfaces colonized with heavy sporulation of *A. flavus* and JL 24 (red) was an *A. flavus* colonization susceptible control.

### 3.4. Relationship between Aflatoxin Contaminations with Total Polyphenol, Pod Yield, Seed Yield, and Haulm Yield

The aflatoxin content in resistant genotypes such as ICG 1415, ICG 2106, ICG 311, ICG 4684, ICG 4543, and the resistant control 55-437 showed a strong positive correlation with the seed coat total polyphenol content (r^2^ = 0.80; 0.82), pod yield (r^2^ = 0.58; 0.14), seed yield (r^2^ = 0.74; 0.31), and haulm yield (r^2^ = 0.47; 0.39) under WW and WS treatments, respectively (Figure 9 and Figure 10).

## 4. Discussion

This study revealed a wide genotypic variation for all of the studied traits under the two water treatments. The effect of the genotype–water-treatment interaction (G × Wtrt) showed that genotype‘s performance differed under well-watered (WW) and water-stressed (WS) treatments. Indeed, the pod yield (PY), the seed yield (SY), and the haulm yield (HY) were decreased by the intermittent water deficit during the seed-filling phase by up to 19.49%, 27.24% and 22.07%, respectively. As for the immature pod number per plant (IMPN) and the aflatoxin contamination (AC), they increased by up to 57.14% and 67.16% under WS. This indicated that the intermittent drought applied from the beginning of the seed-filling phase to maturity reduced the yield and its components, while it increased the aflatoxin contamination and the immature pod number. This results are similar to previous studies which reported that a water deficit imposed during the 60th to the 85th DAS led to a yield loss of up to 26% [37]. It also corroborates other works that reported a 24% yield reduction during the end of the growing season [20,38]. The top ten drought-tolerant genotypes revealed in this study are ICG 12697, ICG 12879, ICG 2019, ICG 2119, ICG 6703, ICG 4598, ICG 4684, ICG 2106, ICG 12988, and ICGIL 11114. These genotypes showed high yields under WS. Four of these genotypes, ICG 12879, ICG 2019, ICG 2106, and ICGIL 11114, belong to the group G1 that showed the highest performance of the yield production under water deficit. The six remaining genotypes belong to the group G2 that showed an average performance of the yield production under drought. However, the genotype ICG 12235 presented a particular performance by producing high pod yields and very low seed yields under water-deficit stress.

The tolerance of some genotypes in this study can be explained by their ability to partition dry matter into harvestable yields under limited water supply [39]. Interestingly, this study also revealed five genotypes, among the top ten drought-tolerant genotypes, presenting low aflatoxin contamination under drought. These genotypes are ICG 12697 (2.68 µg kg^−1^), ICG 2119 (0.90 µg kg^−1^), ICG 4598 (3.23 µg kg^−1^), ICG 4684 (1.93 µg kg^−1^), and ICG 2106 (2.60 µg kg^−1^. This indicated that these genotypes maintained high yields under water deficit and kept their seed coat intact which protected the seeds against *A. flavus* invasion and limited aflatoxin contamination. Previous studies reported that high aflatoxin levels are usually found in damaged pods compared to pods with intact shells [40]. Thus, our findings are similar to the previous results which indicated that a high pod yield under drought conditions is related to low seed infection and low aflatoxin contamination [14,41,42].

These last five drought-tolerant and aflatoxin-resistant genotypes belong to the Spanish and Valencia botanical types. This suggested that the peanut Spanish and Valencia botanical types are more resistant to aflatoxin contamination under intermittent drought during the seed-filling phase than the Virginia type. Because of the Virginia type has a long seed-filling phase which exposed it more to the risk of water stress than Spanish and Valencia types. The long seed-filling phase promoted pod damage, seed coat crack, *A. flavus* invasion, and aflatoxin contamination, while the Spanish and Valencia botanical types have a short seed-filling phase. In addition to the top ten drought-tolerant and low AC genotypes under WS, the genotypes ICG 311 (0.3 µg kg^−1^), ICG 14523 (2.71 µg kg^−1^), ICG 4543 (2.80 µg kg^−1^), ICG 5195 (2.83 µg kg^−1^), ICG 6813 (3.51 µg kg^−1^), and ICG 1415 (3.51 µg kg^−1^) showed a low aflatoxin content. Indeed, the water deficit led to the identification of the genotypes that have a specific tolerance to water deficit and keep the seed coat intact to protect the cotyledons against *A. flavus* invasion and aflatoxin contamination.

This suggests that water-deficit injuries to the pods and testa enable the fungus to enter and infect the kernels [17,40,43]. Recent studies have reported similar conclusions for the low aflatoxin content in the seeds of these genotypes under field conditions [32] and under different levels of drought and temperature [44]. Furthermore, this study also revealed the genotypes with a low aflatoxin content (4 µg kg^-1^) for human consumption based on the European Union commission standard [45]. Others authors have also reported the same conclusion for drought-tolerant genotypes having resistance to aflatoxin contamination [20,46].

In this study, the significant differences in the seed coat total polyphenol (%TPP) and the incidence of *A. flavus* colonization among genotypes could be due to the seed coat color. These results revealed that the seed coat total polyphenol content reduced more *A. flavus* colonization according to the seed coat color than to the quantity of the total polyphenol contained in the seed coat. However, the variations in the incidence of *A. flavus* colonization and the percentage of the total polyphenol (%TPP) were observed mainly in the genotypes ICG 311 (19%; 53.07%), ICG 2119 (100%; 52.14%), ICG 4684 (75%; 45.97%), ICG 14523 (43%; 51.61%), ICG 4543 (28%; 47.55%), ICG 5195 (100%; 52.36%), ICG 6813 (46%; 46.53%), and ICG 1415 (100%; 43.68%), respectively. These results show that the percentage of the seed coat total polyphenol influenced aflatoxin contamination. The result is in agreement with the previous studies which reported that other phenolic contents and the cell wall have an effect on the total polyphenol, inhibiting pre-harvest aflatoxin contamination [24,30]. Thus, the pink small grain such as that of 55-437 which showed a low incidence of *A. flavus* colonization (21%) and a low percentage of the seed coat total polyphenol (49%) was more resistant to *A. flavus* colonization than the dark big grain such as ICG 12235 that presented 85% of the incidence of *A. flavus* colonization and 56.67% of the seed coat total polyphenol. Furthermore, aflatoxin contamination under the intermittent water deficit was significantly associated with the seed coat total polyphenol content and the seed size. These results corroborate those of previous studies which reported that the difference in mycelial growth surface coverage can be explained by the differences in the physical and chemical features of the seed coat, pod-shell thickness, and reticulation [10,30,47,48]. The peanut seed coat is composed of multiple cell wall layers, and peanut varieties differ in their composition of flavonoids and tannins which ultimately give different colors to the peanut seed coat [24,30,49].

In this study, the drought-tolerant genotypes ICG 1415, ICG 2106, ICG 311, ICG 4684, and ICG 4543 revealed a significant relationship between aflatoxin resistance and seed coat total polyphenol under two water treatments (r^2^ = 0.80; r^2^ = 0.82).The seed coat total polyphenol of the drought tolerant genotypes showed an association with the pod yield (r^2^ = 0.58; r^2^ = 0.14), the seed yield (r^2^ = 0.74; r^2^ = 0.31), and the haulm yield (r^2^ = 0.47; r^2^ = 0.39) y, under WW and WS respectively. This indicated that these drought-tolerant genotypes kept their seed coat intact and minimized aflatoxin contamination under the intermittent water deficit. Our findings are similar to previous studies reporting that the existence of the seed coat resistance was a logical assumption, considering that seeds with damaged testa are more easily and rapidly invaded by fungus than those with intact testa, and that colored testa conferred greater resistance to *A. flavus* invasion than white or variegated testa [50,51].

## 5. Conclusions

Our findings showed that an intermittent water deficit during the seed-filling phase significantly decreased peanut yields and increased aflatoxin contamination. The peanut genotypes with a pink color of the seed coat belonging to the Spanish and Valencia botanical type are more resistant against *A. flavus* invasion and aflatoxin contamination than the Virginia botanical type under an intermittent water deficit because of their short seed-filling phase. From this study, the genotypes ICG 12697, ICG 2119, ICG 4684, ICG 2106, ICG 311, ICG 4543, ICG 5195, and ICG 1415 showed the best performance for drought-tolerance and low aflatoxin content. These genotypes can be used in a breeding program to select the varieties that combine drought-tolerance and minimize aflatoxin contamination in the semi-arid tropics.

## Figures and Tables

**Figure 1 jof-10-00850-f001:**
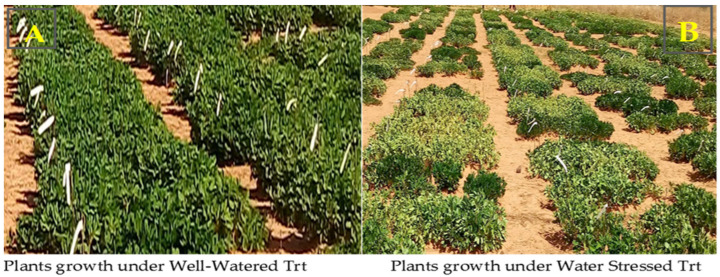
Pictures of plants growing taken at 70 days after sowing under two water treatments. Trt = treatment. (**A**) the plots showing the plants’ growth under well-watered treatment; (**B**) the plots showing the plants’ growth under water-stressed treatment.

**Figure 2 jof-10-00850-f002:**
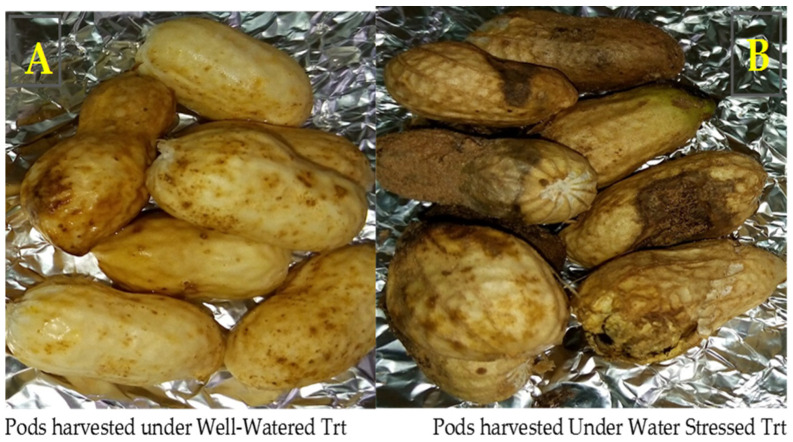
Pods after harvest under well-watered (**A**) and water-stressed (**B**) treatments. Trt = treatment.

**Figure 3 jof-10-00850-f003:**
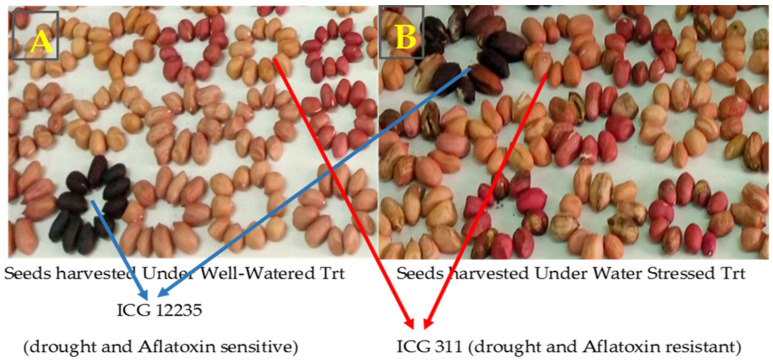
Types of seeds obtained under well-watered treatment and stressed treatment. Blue dash shows the seed coat of the drought-sensitive genotype ICG 12235 under well-watered (**A**) and water-stressed treatments (**B**). Red dash shows the seed coat of the drought-tolerant genotype ICG 311 under well-watered (**A**) and water-stressed treatments. (**B**) Trt = treatment. (**A**) seeds under well-watered treatment; (**B**) seeds under water-stressed treatment.

**Figure 4 jof-10-00850-f004:**
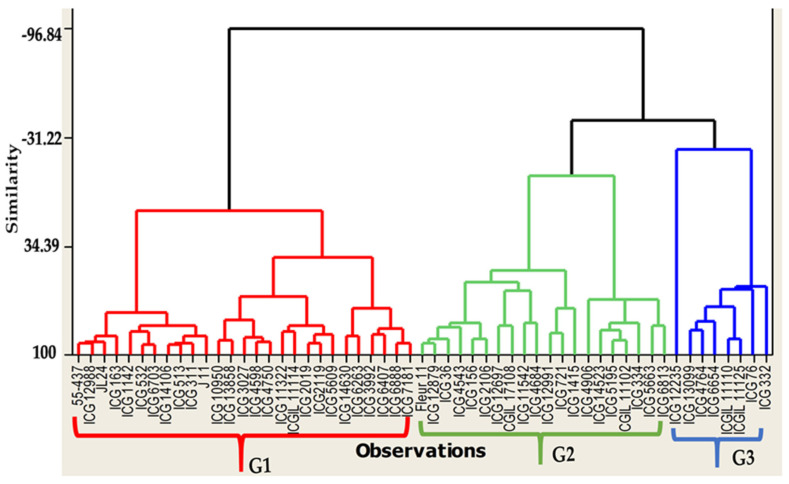
Classification of the fifty-five genotypes in 3 groups (G1, G2, and G3) of performance under well-watered treatment. G1 = high performance, G2 = low performance, and G3 = average performance.

**Figure 5 jof-10-00850-f005:**
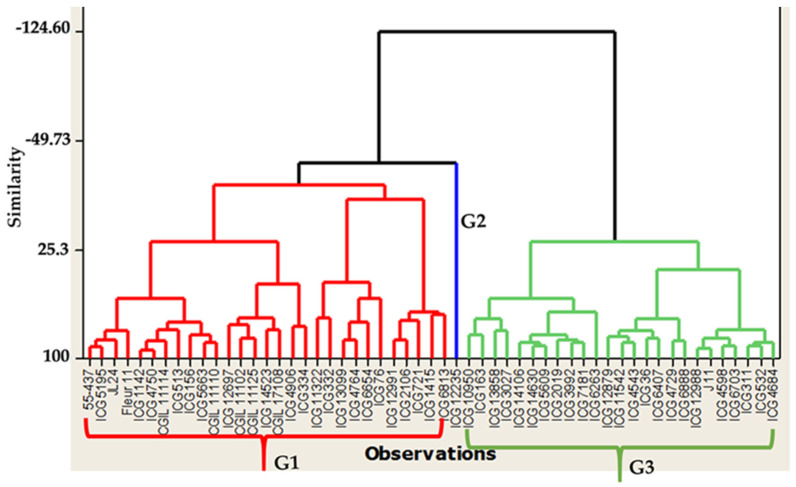
Classification of the fifty-five genotypes in 3 groups (G1, G2, and G3) of performance under water-stressed treatment. G1 = high performance, G2 = low performance, and G3 = average performance.

**Figure 6 jof-10-00850-f006:**
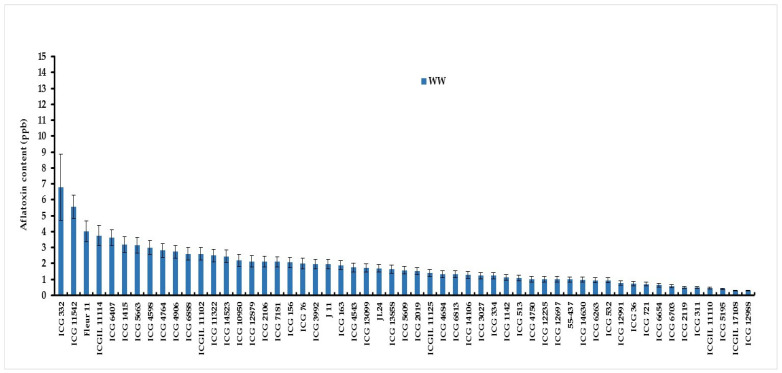
Aflatoxin content (µg kg^−1^) variation in fifty-five peanut genotypes under well-watered (WW) treatment.

**Figure 7 jof-10-00850-f007:**
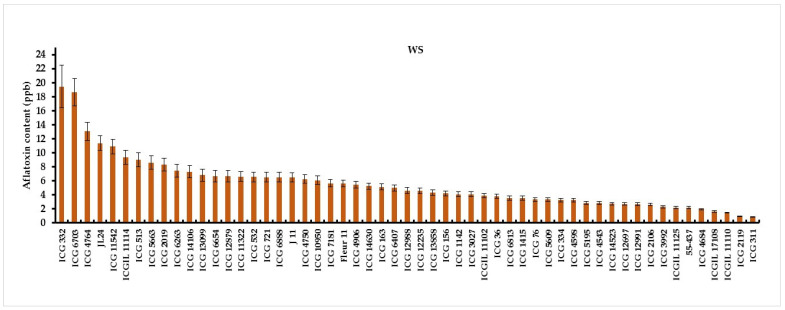
Aflatoxin content (µg kg^−1^) variation in fifty-five peanut genotypes under water-stressed (WS) treatment.

**Figure 8 jof-10-00850-f008:**
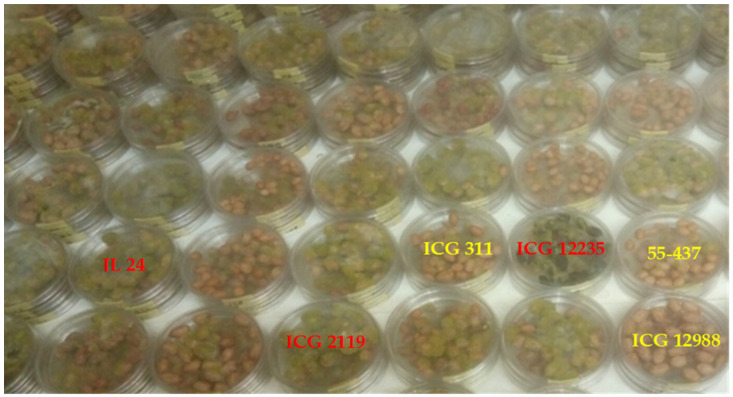
In vitro seed colonization test of fifty-five genotypes at 7 days (32 °C) after *A. flavus* spores’ inoculation.

**Figure 9 jof-10-00850-f009:**
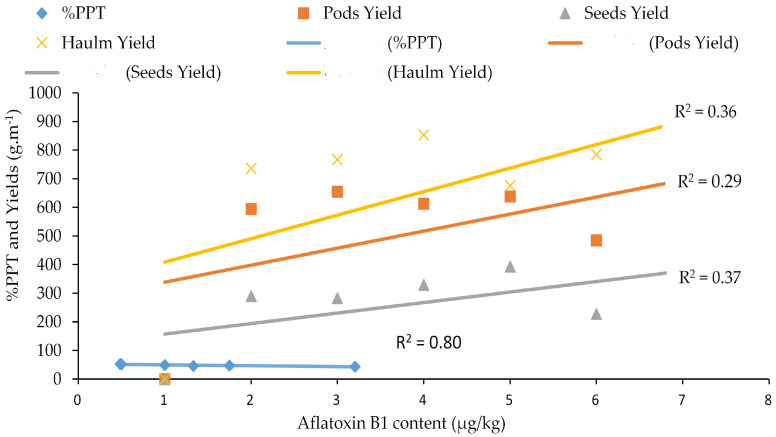
Relationship between aflatoxin contamination and seed coat total polyphenol content, pod yield, seed yield, haulm yield under well-watered conditions (WW). %TPP (
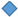
) = percentage of total polyphenol; PY (
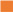
) = pod yield, SY (
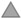
) = seed yield, and HY (
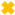
) = haulm yield. (

) = linear %TPP curve, (

) = linear pod yield curve, (

) = linear seed yield curve, and (

) = linear haulm yield curve.

**Figure 10 jof-10-00850-f010:**
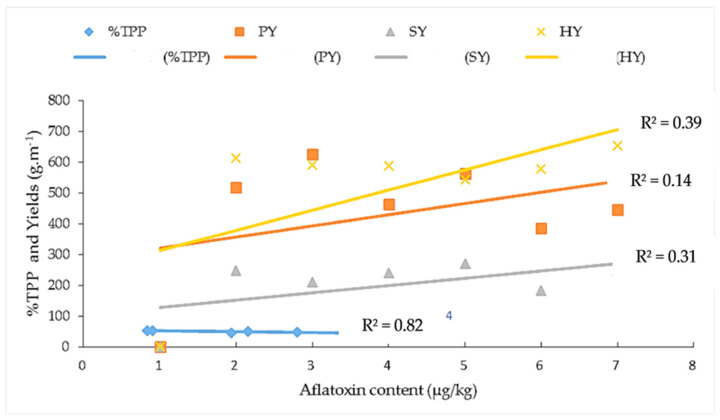
Relationship between aflatoxin contamination and seed coat total polyphenol content, pod yield, seed yield, haulm yield under stressed conditions (WS). %TPP (
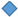
) = percentage of total polyphenol; PY (
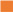
) = pod yield, SY (
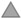
) = seed yield, and HY (
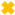
) = haulm yield. (

) = linear %TPP curve, (

) = linear pod yield curve, (

) = linear seed yield curve, and (

) = linear haulm yield curve.

**Table 1 jof-10-00850-t001:** Results of variance analysis (F value), F (probability), means, LSD (least significant differences of means at the 5% level) of yield and its components. Genotype (G), water treatment (Wtrt), and genotype–water-treatment interaction (G × Wtrt) effects were tested. Wtrt = water treatment; WW = well-watered; WS = water-stressed; Vr. = variance; PNP = pod number per plant^−1^; SNP = seed number per plant^−1^; IMPN = immature pod number per plant^−1^; HY = haulm yield; PY = pod yield; SY = seed yield.

Sources of Variance	PNP		IMPN		SNP		HY (gm^−2^)		PY (gm^−2^)		SY (gm^−2^)	
	WW	WS	WW	WS	WW	WS	WW	WS	WW	WS	WW	WS
F Vr.Value	31.62	31.75	12	3.25	14.23	12	8.25	4.24	14	12	7.24	9.16
G (F prob)	<0.001	<0.001	<0.001	<0.001	<0.001	<0.001	<0.001	<0.001	<0.001	<0.001	<0.001	<0.001
Wtrt (F prob)	<0.001	<0.001	<0.001	<0.001	<0.001	<0.001	<0.001	<0.001	<0.001	<0.001	<0.001	<0.001
G × (Wtrt)	<0.001		<0.001		<0.001		0.015		0.027		0.020	

Significance at the 0.05, 0.01, and 0.001 level.

**Table 2 jof-10-00850-t002:** Means variation in yield and its components under well-watered (WW) and water-stressed (WS) treatments and the percentages of decrease and increase.

Traits	Water Treatments		WS Negative Effect (%)
	WW	WS	
Pod number per plant	102 ± 31	80 ± 25	21.92
Immature pods per plant (increasing)	9 ± 5	21 ± 10	57.14
Seed number per plant	123 ± 31	89 ± 27	27.36
Pod yield (gm^−2^)	678.54 ± 203.51	546.15 ± 163.5	19.43
Seed yield (gm^−2^)	316.67 ± 90.26	230.37 ± 69.3	27.24
Haulm yield (gm^−2^)	828.53 ± 85.30	645.57 ± 150.13	22.07

**Table 3 jof-10-00850-t003:** Analysis of variance (variance value), F (probability), means, LSD (least significant differences of means 5% level) of aflatoxin content under well-watered and stressed treatments. Variance value = Vr. Value.

	Aflatoxin B1	
Genotypes	WW	WS
Means	1.81	5.51
LSD	1.14	1.49
Vr.Value	9.55	56
F (Prob)	<0.001	<0.001
G × W(Trt)	<0.001	

Significance at the 0.05, 0.01, and 0.001 level.

**Table 4 jof-10-00850-t004:** Results of variance analysis (variance value), F (probability), means, LSD (least significant differences of means at the 5% level) of the percentage of seed coat total polyphenol (%TPP) and the percentage of *Aspergillus flavus* colonization (AFC). Variance value = Vr. Value.

Genotypes	%TPP	%AFC
Means	50.50	85
LSD	2.54	16.63
Vr. Value	19.38	4.12
F (Prob)	<0.001	<0.001

Significance at the 0.05, 0.01, and 0.001 level.

**Table 5 jof-10-00850-t005:** Fifty-five genotypes ranked according to the percentage of *A. flavus* colonization incidence and the percentage of seed coat total polyphenol content. CI = colonization incidence; percentage of *A. flavus* colonization = %AFC; percentage of total polyphenol = %TPP.

	Genotypes	%AFC	%TPP	Genotypes	%AFC	%TPP	Genotypes	%AFC	%TPP
CI ≤ 25%	ICG 311	19	53.08	55-437	21	49	ICG 12988	18	52.1
CI ˃ 25% ≤ 50%	ICG 11322	50	50.43	ICG 14523	43	51.61	ICG 1415	43	51.61
	ICG 11542	28	43.90	ICG 163	45	47.38	ICG 3027	83	57.50
	ICG 4543	28	47.55	ICG 76	42	52.57	J 11	40	55.16
CI ˃ 50%	Fleur 11	86	47.90	ICG 10950	91	56.05	ICG 1142	100	50.32
	ICG 12235	85	56.67	ICG 12697	91	51.47	ICG 12879	100	45.50
	ICG 12991	81	48.75	ICG 13099	66	50.35	ICG 4906	83	46.86
	ICG 13858	91	53.27	ICG 14106	100	48.76	ICG 1415	100	43.68
	ICG 14630	81	56.95	ICG 14630	81	56.95	ICG 156	90	51.86
	ICG 2019	100	52.34	ICG 2106	83	48.63	ICG 332	100	53.22
	ICG 334	66	50.76	ICG 36	66	49.14	ICG 3992	83	53.17
	ICG 4598	100	48.16	ICG 4684	75	45.97	ICG 2119	100	52.41
	ICG 4750	65	47.38	ICG 4764	70	50.23	ICG 513	100	53.69
	ICG 5195	100	52.36	ICG 532	100	42.74	ICG 5609	83	43.74
	ICG 5663	81	49.36	ICG 6263	91	54.01	ICG 6407	61	52.14
	ICG 6654	91	48.31	ICG 6703	91	46.32	ICG 6813	46	46.53
	ICG 6888	83	46.09	ICG 7181	100	45.92	ICG 721	91	50.20
	ICGIL 11102	100	53.09	ICGIL 11110	75	51.29	ICGIL 11114	75	54.34
	ICGIL 11125	61	56.30	ICGIL 17108	91	62.96	JL24	100	47.93

## Data Availability

The original contributions presented in the study are included in the article/Appendix A, further inquiries can be directed to the corresponding author.

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
