# Peer review of "Aflatoxin B1 Contamination Association with the Seed Coat Biochemical Marker Polyphenol in Peanuts Under Intermittent Drought"

_jof, 2024, doi:10.3390/jof10120850_

Round 1
Reviewer 1 Report
Aflatoxin contamination increases as the severity of drought stress increase in agricultural products, especially peanuts. Identifying drought-tolerant peanut genotypes resistance to Aspergillus flavus may aid in minimizing aflatoxin contamination. The findings of this manuscript might be useful in selecting the drought tolerant genotype of peanut to minimize aflatoxin contamination. But there are many deficiencies found in this manuscript, they should be carefully revised thoroughly.
1. The references are too old, too few references in the past five years.
2. Is “pthe olyphenol” in Line21 correct?
3. Is “diff--erential” in Line70 correct?
4. The title is no match the study content perfectly, it should be changed to be “Aflatoxin Contamination Association with the Seed Coat Total Polyphenol in Peanut under Intermittent Drought”, “Aflatoxin Contamination Association with Seed Coat Biochemical Marker- Polyphenol in Peanut under Intermittent Drought”, or other more fitted one.
5. The content in Line73 and Line 57 are simple repeated.
6. There are lots of format problems throughout the manuscript, some examples are given below:
Line 95 “45km” should be “45 km”; Line 114 “2m x 1m” should be “2 m x 1 m”; Line 187 “table2” should be “Table 2”; Line 203 “Tablee3” should be “Table 3”; Line 241 “(0.48)” should be “(0.48 ppb)”
7. What are “TPP” and “AFC” in the manuscript? They should be clearly explained when appeared first time in the context.
8. Pictures of peanuts pods, seeds and plants should be presented, showing the significant difference of pod, seed, and so on, of each genotype under corresponding watering treatment.
Author Response
Response to Reviewer 1 Comments
|
|||||||||||||
1. Summary |
|
|
|||||||||||
Thank you very much for taking the time to review this manuscript. Please find the detailed responses below and the corresponding revisions/corrections highlighted/in track changes in the re-submitted files.
|
|||||||||||||
2. Questions for General Evaluation |
Reviewer’s Evaluation |
Response and Revisions |
|||||||||||
Does the introduction provide sufficient background and include all relevant references? |
Yes/Can be improved/Must be improved/Not applicable |
[Please give your response if necessary. Or you can also give your corresponding response in the point-by-point response letter. The same as below] |
|||||||||||
Are all the cited references relevant to the research? |
Yes/Can be improved/Must be improved/Not applicable |
|
|||||||||||
Is the research design appropriate? |
Yes/Can be improved/Must be improved/Not applicable |
|
|||||||||||
Are the methods adequately described? |
Yes/Can be improved/Must be improved/Not applicable |
|
|||||||||||
Are the results clearly presented? |
Yes/Can be improved/Must be improved/Not applicable |
|
|||||||||||
Are the conclusions supported by the results? |
Yes/Can be improved/Must be improved/Not applicable |
|
|||||||||||
3. Point-by-point response to Comments and Suggestions for Authors |
|||||||||||||
Comments 1: [The references are too old, too few references in the past five years.] |
|||||||||||||
Response 1: [We used these references because they are perfectly match with our study. Some of these references are belonging to the past five years] Thank you for pointing this out. We agree with this comment. |
|||||||||||||
Comments 2: [Is “pthe olyphenol” in Line21 correct?] |
|||||||||||||
Response 2: Agree. We have admitted this mistake; accordingly, we revised it to become polyphenol to emphasize this point. This correction is located in page1, paragraph1, line22.
|
|||||||||||||
4. Response to Comments on the Quality of English Language |
|||||||||||||
Point 1: |
|||||||||||||
Response 1: (in red) |
|||||||||||||
5. Additional clarifications |
|||||||||||||
[Here, mention any other clarifications you would like to provide to the journal editor/reviewer.] |
Reviewer 2 Report
Aflatoxin contamination increases as the severity of drought stress increase in peanut. This study presents the drought-tolerant genotypes and resistant to aflatoxin contamination. These findings can be used to select the genotype that combine drought tolerance and minimizing aflatoxin contamination
The following are some specific comments:
1. For the experiments, several cultivars were selected, why selected these, what is the principles of your choice?
2. Aflatoxins contain several types including B, G, M.., in this study what you mean? Which kind? Please specify.
3. For the detection, ELISA method has been used, why not used the HPLC, it is more precise than TCL or ELISA.
Aflatoxin contamination increases as the severity of drought stress increase in peanut. This study presents the drought-tolerant genotypes and resistant to aflatoxin contamination. These findings can be used to select the genotype that combine drought tolerance and minimizing aflatoxin contamination
The following are some specific comments:
1. For the experiments, several cultivars were selected, why selected these, what is the principles of your choice?
2. Aflatoxins contain several types including B, G, M.., in this study what you mean? Which kind? Please specify.
3. For the detection, ELISA method has been used, why not used the HPLC, it is more precise than TCL or ELISA.
4. some error like the space between in Line 53, to and drought? What does “pthe olyphenol” mean in Line 21?
Author Response
For research article
Response to Reviewer 2 Comments
|
|||||||
1. Summary |
|
|
|||||
Thank you very much for taking the time to review this manuscript. Please find the detailed responses below and the corresponding revisions/corrections highlighted/in track changes in the re-submitted files. |
|||||||
2. Questions for General Evaluation |
Reviewer’s Evaluation |
Response and Revisions |
|||||
Does the introduction provide sufficient background and include all relevant references? |
Yes/Can be improved/Must be improved/Not applicable |
[Please give your response if necessary. Or you can also give your corresponding response in the point-by-point response letter. The same as below] |
|||||
Are all the cited references relevant to the research? |
Yes/Can be improved/Must be improved/Not applicable |
|
|||||
Is the research design appropriate? |
Yes/Can be improved/Must be improved/Not applicable |
|
|||||
Are the methods adequately described? |
Yes/Can be improved/Must be improved/Not applicable |
|
|||||
Are the results clearly presented? |
Yes/Can be improved/Must be improved/Not applicable |
|
|||||
Are the conclusions supported by the results? |
Yes/Can be improved/Must be improved/Not applicable |
|
|||||
3. Point-by-point response to Comments and Suggestions for Authors |
|||||||
Comments 1: [. For the experiments, several cultivars were selected, why selected these, what is the principles of your choice?] |
|||||||
Response 1: [ We selected all of cultivars because of their contrasting for aflatoxin content reported previsiouly by Waliyar et al., 2016 without the six (6) ICGILs that are news cultivars , and the principles of our choice is seed coat color diversity.] Thank you for pointing this out. We agree with this comment. Therefore, we have revised the part and complete with this information in the materiel section in the line 89 to 90. |
|||||||
Comments 2: [. Aflatoxins contain several types including B, G, M.., in this study what you mean? Which kind? Please specify.] |
|||||||
Response 2: Agree. We have accepted these corrections; accordingly, we revised to emphasize this point. In this study, we mean aflatoxin B1 (AFB1) only. We corrected this in the manuscript since the first appearing in Line 2 and Line 15.
|
|||||||
Point 1: |
|||||||
Response 1: (in red) |
|||||||
5. Additional clarifications |
|||||||
[Here, mention any other clarifications you would like to provide to the journal editor/reviewer.] |
Reviewer 3 Report
The manuscript describes an interesting experiment on the drought tolerance and some seed coat treatments associated with Aspegillus flavus resistance and aflatoxin contamination. The experimental design seems correct and the results may be of interest to the readers of Journal of Fungi - after modification, as detailed below.
-L.16-17: The objectives of the study should be clearly presented in the Abstract.
-L.68-69: Check this sentence for clarity. In fact, the English should be carefully revised througout the text.
-L.128-131: Rewrite those lines to indicate what are the yield components evaluated in the study.
-L.165-172: Section "2.7 Genotypes Aflatoxin content quantification": Details of the ELISA (not "ELIZA") method should be fully described, including the brand of kits used, limits of detection and quantification, recovery and precision. Just mentioning the reference [26] is not enough for accepting the validity of results on aflatoxin contents in the samples analyzed.
-L.173: You mean "Statistical analysis"?
-L.184-220: Do not repeat information already displayed in tables. So, those lines should be greatly summarized to only highlight the main findings, not all the detailed results.
-In Tables 1 and 2, what are the meanings of "WW" and "WS"?
-Table 3: Present this table as supplementary material.
-L.240 (and througout the text): "ppb" is not an international standard unit. So, use "micrograms per kilogram" or an equivalent unit.
-Figures 3 and 4: Some parts in these figures are not in English.
Author Response
For research article
Response to Reviewer 3 Comments
|
||||||||||||||||||
1. Summary |
|
|
||||||||||||||||
Thank you very much for taking the time to review this manuscript. Please find the detailed responses below and the corresponding revisions/corrections highlighted/in track changes in the re-submitted files.
|
||||||||||||||||||
2. Questions for General Evaluation |
Reviewer’s Evaluation |
Response and Revisions |
||||||||||||||||
Does the introduction provide sufficient background and include all relevant references? |
Yes/Can be improved/Must be improved/Not applicable |
[Please give your response if necessary. Or you can also give your corresponding response in the point-by-point response letter. The same as below] |
||||||||||||||||
Are all the cited references relevant to the research? |
Yes/Can be improved/Must be improved/Not applicable |
|
||||||||||||||||
Is the research design appropriate? |
Yes/Can be improved/Must be improved/Not applicable |
|
||||||||||||||||
Are the methods adequately described? |
Yes/Can be improved/Must be improved/Not applicable |
|
||||||||||||||||
Are the results clearly presented? |
Yes/Can be improved/Must be improved/Not applicable |
|
||||||||||||||||
Are the conclusions supported by the results? |
Yes/Can be improved/Must be improved/Not applicable |
|
||||||||||||||||
3. Point-by-point response to Comments and Suggestions for Authors |
||||||||||||||||||
Comments 1: [-L.16-17: The objectives of the study should be clearly presented in the Abstract.] |
||||||||||||||||||
Response 1: [We accepted to revise the objectives of the study in Abstract, as the goal is to identify the drought tolerant genotypes with seed coat biochemical resistance to A.flavus and aflatoxin contamination Thank you for pointing this out. We agree with this comment. Therefore, we have modified the objectives in the manuscript in the Lines 17-19.] |
||||||||||||||||||
Comments 2: [-L.68-69: Check this sentence for clarity. In fact, the English should be carefully revised througout the text] |
||||||||||||||||||
Response 2: Agree. We have accepted to clarify the sentence in Line 68 and Line 69, accordingly, we revised these sentences to emphasize this point. The sentences be become recent study reported that the wide variation of the incidence and the severity of A. flavus colonisation, and the aflatoxin content between the genotypes can be biochemical compounds differential variability in the tested seeds. The revision can be found in Line 69-72 in the manuscript.
|
||||||||||||||||||
4. Response to Comments on the Quality of English Language |
||||||||||||||||||
Point 1: |
||||||||||||||||||
Response 1: (in red) |
||||||||||||||||||
5. Additional clarifications |
||||||||||||||||||
[Here, mention any other clarifications you would like to provide to the journal editor/reviewer.] |

Round 2
Reviewer 1 Report
Aflatoxin contamination increases as the severity of drought stress increase in agricultural products, especially peanuts. Identifying drought-tolerant peanut genotypes resistance to Aspergillus flavus may aid in minimizing aflatoxin contamination. The findings of this manuscript might be useful in selecting the drought tolerant genotype of peanut to minimize aflatoxin contamination. But there are still several problems found in this manuscript, even in the revised part. All the problems should be carefully revised.
1. The resolution of both pictures provided in the Figure 1 (Line 113) are not clear enough, and there should be some problems happened in the legends for both panels, both of them are “Plants growing under well watered”.
2. Pictures of peanuts pods or seeds should be presented and compared.
Author Response
Comments 1: [The resolution of both pictures provided in the Figure 1 (Line 113) are not clear enough, and there should be some problems happened in the legends for both panels, both of them are “Plants growing under well watered”]
Response 1: [We Improved the resolution of both pictures provided in the Figure 1 Line 113. The Figure 1 becomes Figure 3 in the section results in Lines 278, 279 in the manuscript] Thank you for pointing this out. We agree with this comment. Therefore, we have transferred this Figure in the section results to show clearly the symptom of water deficit on the plants growing in the field. Because this picture explained our results. We brought the references in the past five years in the revised manuscript. These references can be found in the section introduction in the Line 40 references 3; Line 42 reference 4; Line 43 references 5 and 6; Line 46 reference 11; Line 47 reference 14; Line 55 reference 14; Line 65 reference 24.
Comments 2: [ Pictures of peanuts pods or seeds should be presented and compared] |
Response 2: We have brought the pictures of peanuts pods and seeds in the revised manuscript. This change can be found in the section results in Lines 280,…289 Figure 4 and Figure 5. These Figures showed the pictures of the pods and seeds harvested under well watered and water stress treatments. These Figures describe clearly our results by showing the damages caused by intermittent water deficit applied during the seed-filling phase. Agree. We have accepted this pertinent suggestion; accordingly, we revised the manuscript to emphasize this point.
|
Reviewer 2 Report
The authors have made sufficient modifications, it can be accepted.
Sufficient changes have been made.
Author Response
Comments 1: [The authors have made sufficient modifications, it can be accepted.] |
Response 1: [The revised manuscript has been improved again] Thank you for pointing this out. We agree with this comment. Therefore, we have brought the pictures of the pods, the seeds harvested under well-watered treatment, and water stressed treatment. These pictures described clearly the damages caused by the intermittent water deficit. The pictures can be found in the section results in Lines 278,….289. The picture of in-vitro seed colonization test has brought also in the revised manuscript in the section results in Lines 341 ...346. We brought also the references in the past five years in the revised manuscript. These references can be found in the section introduction in the Line 40 references 3; Line 42 reference 4; Line 43 references 5 and 6; Line 46 reference 11; Line 47 reference 14; Line 55 reference 14; Line 65 reference 24. |
Comments 2: [Sufficient changes have been made] |
Response 2: Agree. We have made some changes again in the revised manuscript; accordingly, We changed and improved the picture resolution of Figure 1 Line 113. This figure is removed to the results section to be came Figure 3 which described clearly our results to show the damage caused by the intermittent water deficit in Line 278 to emphasize this point.
|

Round 3
Reviewer 1 Report
Aflatoxin contamination increases as the severity of drought stress increase in agricultural products, especially peanuts. Identifying drought-tolerant peanut genotypes resistance to Aspergillus flavus may aid in minimizing aflatoxin contamination. The findings of this manuscript might be useful in selecting the drought tolerant genotype of peanut to minimize aflatoxin contamination. But minor correction of this manuscript is still necessary.
Example: Line 221 “table” should be “Table”
Author Response
Comments 1: [ Line 221 “table” should be “Table”]
Response 1: [We have accepted to correct this mistake] Thank you for pointing this out. We agree with this comment. Therefore, we have revised the writing of word table to Table in the manuscript in Line 221.
